# The Transition to Adulthood for Youth Living with Rare Diseases

**DOI:** 10.3390/children9050710

**Published:** 2022-05-12

**Authors:** Melanie Sandquist, TjaMeika Davenport, Jana Monaco, Maureen E. Lyon

**Affiliations:** 1Children’s National Hospital, Center for Translational Research, Washington, DC 20010, USA; msandquist@childrensnational.org; 2Milken School of Public Health, George Washington University, Washington, DC 20010, USA; 3Children’s National Hospital, Goldberg Center for Community Pediatric Health, Washington, DC 20010, USA; tdavenpo@childrensnational.org; 4Children’s National Hospital, National Patient and Family Advisory Council, Washington, DC 20010, USA; jana.monaco@verizon.net; 5School of Medicine and Health Sciences, George Washington University, Washington, DC 20052, USA

**Keywords:** pediatric to adult transition, rare disease, special needs, interventions, care coordination, transition readiness

## Abstract

More children with rare diseases survive into adulthood. The transition period to adult healthcare presents many challenges for pediatric rare diseases. Few adolescents or their families receive any transitional support for the transition to adult healthcare or for their maturing psychosocial needs. Understanding the challenges in the transition process is critical to ensure that interventions designed to improve the transition are holistic and meet the needs of the youth and their families. Few transition programs are in place to meet the needs of those youth with rare diseases who cannot participate in medical decision making or who live independently because of severe disabilities and comorbidities. We searched the literature on preparation and outcomes for youth living with rare diseases in PubMed, CINAHL, and PsychInfo, excluding publications before 2010. The results revealed seven studies specific to rare diseases, special needs, or chronic conditions. Next, we discussed transition with experts in the field, GotTransition.org, and citation chaining, yielding a total of 14 sources. The barriers and challenges to transition were identified. Articles discussing solutions and interventions for transition in medically complex children were categorized care coordination or transition readiness. A large portion of children with rare disease are underserved and experience health disparities in transition.

## 1. Introduction

As modern medical practices improve, an increasing number of children with rare diseases are surviving into adulthood [1,2,3]. A 2018 study conducted in Italy estimated that 9.2% of adults in the healthcare registry were transitioned with rare diseases from pediatric institutions. The transition period presents a challenge to many young adults with diseases that are widely considered a pediatric concern [2]. Many receive inadequate care or are lost completely in the gap between pediatric and adult care providers. The National Survey of Children’s Health found that only 18.4% of adolescents in 2019 received any transitional support [4].

TjaMeika Davenport, a Parent Navigator at Children’s National Hospital and a community advisory board member for Got Transition, described the transition for children with complex care needs as one of the most difficult challenges to navigate as a young adult in the US healthcare system. She explained that the transition is uniquely difficult to counter because it is driven by a variety of factors, including the complexity of their care, differing specialist recommendations, and the adult providers’ lack of experience with rare pediatric diseases, among other factors. She detailed several efforts she and her team make to improve transition for adolescent patients and their families at Children’s National, including the parent navigator program and the Got Transition National Resource Center. One such support was a “warm-handoff” strategy, in which patients met with both pediatric primary care or specialty providers they were transitioning away from and the adult specialist they were transitioning towards, creating a support network for both patients and providers. While largely effective, the warm hand-off strategy is expensive and unlikely to be covered by insurance companies without proof of concept, which makes widespread implementation nearly impossible in a privatized healthcare system, such as that in the United States [5,6]. This process also does not address the needs of young adults who may need Emergency Department services that are unable to meet their special needs nor the loss of a pediatrician before finding an adult provider or practice to transition into. Parents, such as Jana Monaco whose young adult son has a rare genetic metabolic disorder, are left with the case management role of facilitating this transition to adult healthcare with little professional support. As efforts to improve transitional healthcare increase, the need for transitional care research is also increasing. This literature review will outline the current state of transition literature on the barriers and solutions faced by adolescents and young adults with rare diseases who require complex care and who have special needs.

## 2. Materials and Methods

The initial search of literature was conducted in PubMed, CINAHL, and PsychInfo. As seen in Figure 1, the keywords “pediatric to adult transitions,” “rare disease,” and “special needs” returned 39 peer-reviewed journal articles. These keywords were selected to define a specific research goal: transition of care in children with non-specific, ultra-rare disease resulting in special needs and/or complex care. The National Organization for Rare Disorders lists over 1200 rare diseases. There are transition models for specific rare conditions, such as whole organ transplant or Cystic Fibrosis [5,6]. While specific rare disease transition models address specific and general transition challenges, including the literature on the transition of each disease is not feasible at this time. For this reason, the literature review was limited to non-specific rare disease and special needs transition.

After the exclusion of articles written before 2010 and in languages other than English, there were 29 results. The results were further refined by including only articles that discuss the transition between pediatric and adult care for children with rare diseases, special needs, or chronic conditions, and excluding articles that were not accessible in the United States. The final number of results was seven. Due to the limited results, we looked to other means of identifying relevant articles. A total of seven additional journal articles were identified through discussion with experts in the field, GotTransition.org, and citation chaining, including 1 unpublished manuscript. In total, 14 sources were selected. The included articles were then identified as contributing to the literature on the barriers to transition and possible solutions to transition barriers.

## 3. Results

### 3.1. Barriers and Challenges to Transition

An article published in *Pediatrics* details a number of barriers adolescents, young adults, and their families face during healthcare transition and why transition care is so necessary [2]. The authors defined transition as more than the simple physical transfer of a patient from one practice or hospital to another, but instead as the designed effort to ensure healthcare independence, preparation, and the completion of this transfer. Barriers to the transition include the loss of ancillary staff common in pediatric settings, healthcare culture differences, and even simply the work involved in transferring medical records. This is exacerbated in countries such as the US where the transfer of insurance coverage is yet another concern [2].

A 2018 systematic review of barriers to transitional care discovered four overarching themes [7]. The most prominent of which was relationships: Patients were reluctant to leave the providers and staff at their pediatric institutions and were slow to build new ones in the adult care setting. Access, trust, and knowledge issues also pervade the transition process [7]. Patients struggle to find adult specialists willing and capable of treating their conditions, and once they do, they have troubling beliefs and expectations of those providers—yet another barrier to successful care. The authors note that while different illness groups experience these challenges at different rates, the themes are common across most patient types [7].

Similar barriers were identified by adult providers engaging in transitional care [2,4]. A 2021 focus group of interdisciplinary adult providers conducted by the same research group found that providers struggle to perform post-transitional care when a patient or family’s beliefs and expectations of adult care do not align with that of the provider, causing distrust and resistance to change. The providers also explained a lack of communication with the patient’s pediatrician limits their ability to treat the patient, especially when they do not have access to medical records and histories [4]. The final theme identified by the focus group was issues related to access and insurance; interdisciplinary care coordination and social work are largely inaccessible in an adult care setting compared to the interconnectedness of pediatric care, a problem that may be better addressed by payor- and system-level intervention.

A qualitative cross-sectional survey of providers who care for young adults with chronic diseases and complex healthcare needs was conducted in 2015 [8]. Researchers coded responses into 5 themes: size of the medical team, access to medical records, time constraints and administrative burden, lack of training and experience in pediatric diseases, and financial constraints [8]. Although it has a more robust study design than a focus group, the paper has its limitations. Similarly, to the focus group, this study has a small sample size. Only 22 providers responded to the survey, limiting its power and generalizability. It is difficult to say if these providers’ perspectives represent the average transitional adult provider.

### 3.2. Solutions

The articles that discuss solutions and interventions for transition of care in medically complex children can be split into two categories: care coordination and transition readiness. This section also includes examples of generalized transition models in Europe.

#### 3.2.1. Care Coordination

The results of a randomized control trial in young adults with chronic illnesses showed that healthcare transition care coordination was effective in improving patients’ perceptions of care in a transitional period [9]. Those who received the care coordination were more than two times as likely to report receiving the care they thought they needed and speak to their providers about their future care than patients in the control group. Although the study design is strong and these results are promising, the study is limited by its use of self-reported measures. Because the authors were not able to use a more empirical measure of care, only the difference in patients’ perceptions can be confirmed.

Another multidisciplinary transition team was designed and implemented at the Children’s Hospital of Philadelphia [10]. The team consulted on 80 cases over a 2-year period. The team identified appropriate referrals for over 70% of these cases and created health summaries for 90%. In a program evaluation, 78% of referring pediatricians felt the program helped them identify adult providers for their patients, and 90% planned to use a Multidisciplinary Intervention Navigation Team (MINT) for future transitions. The evaluators concluded that MINT was a worthwhile program and recommended further funding and implementation of the team. The Adult Care and Transition Team (ACTT), formerly known as MINT, is now an established interdisciplinary consulting service that requires a referral by a provider. The team helps with: (1) creating a transition care plan that includes an updated medical summary; (2) finding adult doctors and nurse practitioners; (3) answering health insurance questions; (4) coordinating care across pediatric and adult hospitals; (5) transferring the medical records to the new provider; (6) finding services and support in the community.

A 2018 review of healthcare transition frameworks found that while there are many aspects to a smooth transition, cooperation between the patient’s pediatric and adult physicians is the most impactful solution [3]. They emphasize that transition preparation should begin long before the transition and that the issues faced by patients are better mitigated by a team of providers who may offer different perspectives and solutions.

#### 3.2.2. Transition Readiness

A cross sectional study of 17,114 adolescents and young adults on transition readiness in 2013 found healthcare systems lacking [11]. While providers are encouraging patients to take charge of their own health, approximately 56% of participants never had a conversation about transition with their providers. In addition, only 35% reported discussions about health insurance. While these statistics are somewhat improved for patients in a medical home (46.3 and 46.5, respectively), improvements are still necessary [11]. The authors recommend implementing transition preparation on a system-wide level, specifically with payor systems. They reference adult care to nursing home transition programs in Medicare that may be adapted and improved upon for the younger generation covered by public insurance [11]. Monetary investments may increase transition preparedness outcomes in and out of the medical home. This review did not address the transition needs of families of children with severe cognitive or motor limitations who will never be able to take charge of their own health.

A first step to improving transition readiness is consistent measurement. A 2014 review of transition preparedness measures found 10 widely used measures with published validity date, 6 of which were disease-specific [12]. While each measure was found to be a valid benchmark of transition preparedness, the measures that included patient and family participation were most likely to identify patients in need of intervention. The authors caution that although seemingly effective, the measures were created for and tested in specific populations and are, therefore, not necessarily generalizable to a larger population [12]. Further evaluations of and improvements upon these measures are necessary to create standardized readiness measures. Standardized measures may help to identify disparities in transition readiness by decreasing misclassification when comparing readiness across demographic groups.

In 2014, data from the National Survey of Children with Special Healthcare Needs and the Survey of Adult Transition of Health were analyzed to discover if young adults with a healthcare plan were more likely to use dental services [13]. Dental care is an important, if often overlooked, aspect of healthcare and an important aspect of a smooth transition. The researchers found that having a healthcare plan before transition was significantly associated (OR 1.11, 95% CI 1.04 to 1.18) with increased utilization of dental service in young adults with special healthcare needs but no functional limitations [13]. As a retrospective cohort study, the results are subject to some minor information bias. The study was also limited by lack of data on the type of dental care sought. Whether the patient was seen at a pediatric or adult dentist may determine the effectiveness of their transition. Patients who had been seeing a general dentist since childhood and required no transition may have also swayed the results. While it is a relatively small magnitude, an 11% increase in the likelihood of dental care suggests that a healthcare plan is an effective strategy for improving outcomes during transition [13].

A Vermont health system designed and piloted a chatbot that strived to encourage teenage patients with special needs to engage in their medical care [14]. The intended improvements included increasing personal knowledge about their conditions, medications, and medical history as well as preparing for appointments, contacting their providers, and understanding their insurance systems. There are technical and medical difficulties in the transition of care in rare diseases. Adult experts in rare diseases frequently do not know the adolescent specific needs of young adults, as is also the case in transitioning healthy adolescents [15]. Thus, transitioning patients may benefit from personal knowledge of their own conditions and complex care needs, especially when consulting with their pediatric provider is not feasible. The small sample of patients had a 97% engagement profile and showed improvements in many of categories, especially taking control of their medication and pharmacy refills [14]. Due to the study’s status as a pilot program, the sample size was small (n = 16), severely limiting its power. The use of testing technology also required all participants to not only own a cell phone but be relatively tech literate. This may introduce selection bias. Another limitation of note was the study’s compensation structure. A total of USD 100 was given to participants who completed the entire study, which may have swayed the engagement statistics, particularly in the young adolescents. The authors conclude that the pilot study showed the chatbot has serious potential to improve transition readiness in youth with special needs but caution against using similar technologies in a vacuum: transition is a complex hurdle that requires multiple solutions [14].

#### 3.2.3. European Models

Two notable transition interventions in Europe include the United Kingdom’s Ready-Steady-Go and Germany’s Medizinische Behandlungszentren für Erwachsene mit geistiger Behinderung oder schweren Mehrfachbehinderungen (MZEB) programs. An analysis of transition interventions in a nationalized healthcare setting may help in differentiating the effectiveness of an intervention from the financial barriers to healthcare transition seen in privatized insurance systems.

The Ready-Steady-Go program is a generalized transition approach that starts with patients at 11 years old in the UK [16]. Patients work with their providers to develop a transition plan where both parties are comfortable. This allows the transition to proceed at an individualized pace specific to the needs of the patient. It empowers young patients to have confidence and control over their own healthcare. The wide implementation and success of this program suggests that care coordination and early transition preparation are effective in an environment with fewer financial barriers to be concerned with.

Across Germany, adults with intellectual disabilities are cared for in MZEBs. There are specialized clinics designed to serve adult patients with complex intellectual needs [17]. Their success in treating adult patients with complex pediatric disorders supports evidence that specified care centers are effective in bridging the gap between pediatric and adult care in nationalized health systems.

### 3.3. Gaps in Research

A single article, published in 2010, addressed race- and ethnicity-based disparities in transition care and support [18]. The authors of this systematic review concluded that there were significant differences between racial and ethnic groups, and further intervention should be applied to help fill gaps in transition care. Despite a publishing date of more than a decade ago, this study is the most recent research on the topic, and there is little reason to believe the disparity has been resolved. Although race and ethnic disparities are primarily limited to the United States, further research in outcome and transitional support disparities is required to develop appropriate solutions and interventions.

We identified no research that addressed sexual health in transition care for individuals living with rare diseases. Challenges for those with severe intellectual and motor limitations include sexual and reproductive health. Ethical and legal challenges surround issues such as hormonal suppression or birth control for persons with rare diseases who are unable to give consent. Moreover, we did not identify any research that addressed the impact on health of guardianship at the age of 18, living at home versus in the community or long-term care setting, or contingency planning as families age and can no longer provide care. One question rarely asked in this context is “What does a good life look like for your child as an adult?” Programs or interventions that address additional transition challenges are likely implemented in the US and Europe without publication as transition research is both time and resource intensive.

Countries such as the United States and Italy [19] have no national transition model for individuals living with rare diseases. Similarly, there is no current consensus on safe and equitable healthcare transition for patients with rare diseases in resource-limited countries [20]. Although programs such as Got Transition^®^ [21] have identified the six core elements of health care transition, many countries still lack adequate resources to address these elements. In Canada [22], France [23], and Ireland [24], progress has been made towards developing such models and guidelines for transition programs.

## 4. Discussion

The transition from pediatric to adult healthcare is difficult, especially for families of young adults with ultra-rare diseases who are unable to participate in healthcare decision making because of severe disabilities and/or medical comorbidities with unique treatments. Young adults and their providers face a variety of barriers, including knowledge and skill-based challenges, distrust and low expectations, and access and financial concerns. The specifics of each rare disease or group of rare diseases contributes to unique transition issues. Furthermore, the rarity of experts in these rare pediatric and genetic diseases among adult healthcare professionals is a barrier to successful transition. Creative solutions have yet to be identified to overcome these barriers, most prominently in care coordination and transition readiness. System-based interventions of care coordination seem to be most effective in ensuring that patients are successfully transferred from one practice to another while transition preparation programs help improve patient skills necessary for success in an adult care setting, such as medical knowledge and self-advocacy. Both angles of support would be beneficial to adolescents and young adults with rare diseases attempting to transition to adult care and should recognized as a necessary part of medical care.

Additionally, although not addressed in any of the research or quality improvement studies, family caregivers could benefit from guidance through the process of establishing guardianship. Family caregivers need to be educated about this process as their severely disabled child reaches adulthood (age 18 years in the United States), for those children unable to live independently and/or unable to participate in medical decision making. There should be a better way to support these families so they do not experience shock, fear, or intimidation that they might lose custody of their adult child as they proceed through the legal process of establishing guardianship in the United States. Alternatively, as children transition to adult healthcare, family caregivers may also decide to explore relinquishing guardianship. Referrals to social work resources should link family caregivers with available resources to provide support for transitioning to group homes or creative alternative living situations.

There is a concerning gap in knowledge about the disparities of transition care by race and ethnicity. A large portion of children with rare diseases are members of marginalized groups that regularly experience healthcare disparities [1]. Research is necessary to identify, understand, and combat gaps in transitional care for children living with rare diseases. A recently completed National Institutes of Health Workshop further identified gaps in healthcare systems and payment models that require an evidence base to support reimbursement models for healthcare transition services that work [25].

## Figures and Tables

**Figure 1 children-09-00710-f001:**
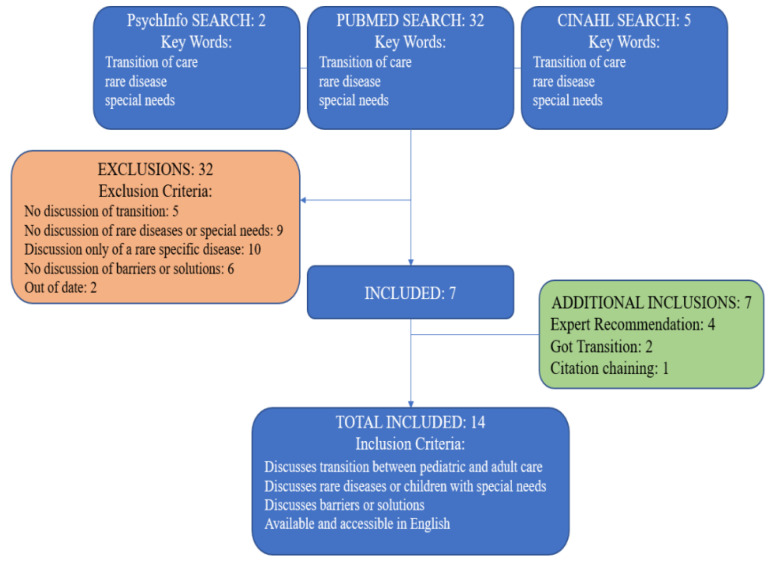
Flow diagram of literature search methods, inclusion, and exclusion criteria.

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
