# Peer review of "The Transition to Adulthood for Youth Living with Rare Diseases"

_children, 2022, doi:10.3390/children9050710_

Round 1
Reviewer 1 Report
Dear all
thank you for this work on a truely important topic.
PLease find below some recommendations for corrections, which I deem necessary.
- You need to be more precise in defining your key-term "rare diseases" there are many different rare diseases which may pose different challenges. At some points you write about "complex care needs" or "special needs" which may be true for some but not all rare conditions.
- This inaccuracy leads to weaknesses in the literature review. There are quite some publications out on specific rare conditions e.g. transplantation, chronic kidney condition, epilepsy and others. Large parts of research focus on a single condition to elaborate on specific needs, which might vary depending on e.g. cognitive abilities.
- In my opinion you are lacking on some models of transition. The UK works a lot with the ready-steady-go programme, which is not disease specific. Germany operates so called MZEB (medical centres for adults with disabilities). These centres bring together various disciplines that are needd in young adults with complex care needs. I am sure that there are some publications out there.
So, in summary, I really appreciate your work on an important topic. However, a more specific definition and a more thorough research of the literature is needed to turn this good approach into a meaningful article
Author Response
Comment 1:
You need to be more precise in defining your key-term "rare diseases" there are many different rare diseases which may pose different challenges. At some points you write about "complex care needs" or "special needs" which may be true for some but not all rare conditions.
Author response:
Thank you for this recommendation. We have chosen to include an expanded explanation of our key words as well as the connection between rare disease and complex care or special needs. Please see included language: ‘These key words were selected to define a specific research goal: transition of care in children with non-specific ultra-rare disease resulting in special needs and/or complex care. The National Organization for Rare Disorders lists over 1200 rare diseases. There are transition models for specific rare conditions, such as whole organ transplant or Cystic Fibrosis. (CITE) While specific rare disease transition models address specific and general transition challenges, including the literature on the transition of each disease is not feasible at this time. For this reason, the literature review was limited to non-specific rare disease and special needs transition.’
Lines 66 to 74.
Comment 2:
This inaccuracy leads to weaknesses in the literature review. There are quite some publications out on specific rare conditions e.g. transplantation, chronic kidney condition, epilepsy and others. Large parts of research focus on a single condition to elaborate on specific needs, which might vary depending on e.g. cognitive abilities.
Author response:
Thank you for highlighting the need for clarification, we have chosen to include an explanation of the parameters of our literature review excluding articles for specific rare disease concerns due to feasibility concerns and the scope of the research question. Please see included language: ‘The National Organization for Rare Disorders lists over 1200 rare diseases. There are transition models for specific rare conditions, such as whole organ transplant or Cystic Fibrosis. (CITE) While specific rare disease transition models address specific and general transition challenges, including the literature on the transition of each disease is not feasible at this time. For this reason, the literature review was limited to non-specific rare disease and special needs transition.’
Lines 69 to 74.
Comment 3:
In my opinion you are lacking on some models of transition. The UK works a lot with the ready-steady-go programme, which is not disease specific. Germany operates so called MZEB (medical centres for adults with disabilities). These centres bring together various disciplines that are needd in young adults with complex care needs. I am sure that there are some publications out there.
Thank you for bringing these programs to our attention, we have chosen to include a section on European models as examples of transition interventions outside the US privatized insurance model. Please see the included language:
‘3.2.3 European Models
Two notable transition interventions in Europe include the United Kingdom’s Ready-Steady-Go and Germany’s Medizinische Behandlungszentren für Erwachsene mit geistiger Behinderung oder schweren Mehrfachbehinderungen (MZEB) programs. Analysis of
transition interventions in a nationalized health care setting may help to differentiate the effectiveness of an intervention from the financial barriers to health care transition seen in privatized insurance systems.
The Ready-Steady-Go programme is a generalized transition approach started with patients at 11 years old in the UK. [14] Patients work with their providers to develop a transition plan that both parties are comfortable with. This allows the transition to go at an individualized pace specific to the needs of the patient. It features empowering young patients to have confidence and control over their own healthcare. The wide implementation and success of this program suggests that care coordination and early transition preparation are effective in an environment with fewer financial barriers to care.
Across Germany adults with intellectual disabilities are cared for in MZEBs. There are specialized clinics designed to serve adult patients with complex intellectual needs. [15] Their success in treating adult patients with complex pediatric disorders supports evidence that specified care centers are effective in bridging the gap between pediatric and adult care in nationalized health systems. ‘
Lines 220 to 238.
Reviewer 2 Report
* General comments
In this literature review on the transition from childhood to adulthood, the authors identified publications that include a scientific research methodology on the transition in rare diseases.
Their purpose is to analyze articles that take into account the needs of young patients and their families.
They insist on the disparity in care and the lack of equity because often patients are unable to express their wishes and needs especially for the most vulnerable and disabled.
This aspect of access to care that must be equitable is well detailed in the article and to the credit of the authors.
This article can be accepted with minor revisions.
* More precise remarks
The main flaw lies in the confusion that is made in access to care and reimbursement terms based on insurance, especially in the United States while in Europe, care for chronic and rare diseases is covered by the health system. This part must be nuanced and clarified
For example, line 85 (and often in the text), it is mentioned reimbursement by medical insurance this is an American issue rather than European where rare and serious diseases are fully covered by the social security.
There are of course financial barriers in some European countries, especially to find adequate facilities and specific pathways for transition but they do not concern patients individually.
This is therefore not an obstacle to the individual transition in Europe.
A distinction must be made between the different care systems in the article.
It would be interesting to know if the barriers to transition are the same in countries where there is a better reimbursement and medical insurance system with overall patient care.
Line 187 to 202
The issue of adolescent medicine, adds to the technical and medical difficulties of the transition in rare diseases. Adult experts in rare diseases frequently do not know the specificities of adolescence, including in healthy subjects.
-Line 202
There are probably initiatives and programmes who have been implemented by numerous teams on transition in rare diseases including in the United States, Europe and other countries but there are not necessarily published or administratively identified.
Scientific research and publications on the transition (mainly in rare diseases) are difficult to carry out.
- Line 204
The problem of inequity of access to care due to race and ethnicity is encountered mainly in the United States.
Line 25
There is a need for a common core of recommendations and measures that are valid for all transition programs.
However, the authors must not neglect the specificities of each rare disease or group of rare diseases with specific transition issue, because of the rarity of experts mainly among adult health professionals.
Conclusion
The authors should keep their original idea and repeat it in the conclusion that any transition program should be implemented (or mandatory) in health systems
Author Response
Comments 1:
The main flaw lies in the confusion that is made in access to care and reimbursement terms based on insurance, especially in the United States while in Europe, care for chronic and rare diseases is covered by the health system. This part must be nuanced and clarified
For example, line 85 (and often in the text), it is mentioned reimbursement by medical insurance this is an American issue rather than European where rare and serious diseases are fully covered by the social security.
There are of course financial barriers in some European countries, especially to find adequate facilities and specific pathways for transition but they do not concern patients individually.
This is therefore not an obstacle to the individual transition in Europe.
A distinction must be made between the different care systems in the article.
It would be interesting to know if the barriers to transition are the same in countries where there is a better reimbursement and medical insurance system with overall patient care.
Author response:
Thank you for highlighting this oversight. We have now included the difference in financial barriers between countries throughout the article as well as in a new European Models section. Please see examples of included text:
‘While largely effective, the warm hand-off strategy is expensive and unlikely to be covered by insurance companies without proof of concept, which makes widespread implementation nearly impossible in a privatized health care system such as that in the United States.’
‘This is exacerbated in countries like the US where transfer of insurance coverage is yet another concern.’
‘Analysis of transition interventions in a nationalized health care setting may help to differentiate the effectiveness of an intervention from the financial barriers to health care transition seen in privatized insurance systems. ’
Lines 51 to 54, Lines 95 to 97, Lines 223 to 226.
Comment 2:
Line 187 to 202
The issue of adolescent medicine, adds to the technical and medical difficulties of the transition in rare diseases. Adult experts in rare diseases frequently do not know the specificities of adolescence, including in healthy subjects.
Author response:
Thank you for this critique. We have included reference to un-published transition teams in the gaps in knowledge section. Please see the included language:
‘Programs or interventions that address additional transition challenges are likely implemented in the US and Europe without publication as transition research is both time and resource intensive.’
Lines 256 to 258.
Comment 3:
-Line 202
There are probably initiatives and programmes who have been implemented by numerous teams on transition in rare diseases including in the United States, Europe and other countries but there are not necessarily published or administratively identified.
Scientific research and publications on the transition (mainly in rare diseases) are difficult to carry out.
Author response:
Thank you for once again highlighting this perspective oversight. We have included the following language to clarify this distinction:
‘Although race and ethnic disparities are primarily limited to the United States, further research in outcome and transitional support disparities is required to develop appropriate solutions and interventions.’
Lines 245 to 247.
Comment 4:
Line 25
There is a need for a common core of recommendations and measures that are valid for all transition programs.
However, the authors must not neglect the specificities of each rare disease or group of rare diseases with specific transition issue, because of the rarity of experts mainly among adult health professionals.
Author response:
Thank you for your recommendation here. We have added language to our methods to explain our choice to limit our research to non-specific rare diseases and special needs transitions. Please see the added language here:
‘The National Organization for Rare Disorders lists over 1200 rare diseases. While there is research for specific rare conditions, such as whole organ transplant or Cystic Fibrosis, including the literature on the transition of each disease is not feasible at this time. For this reason, the literature review
was limited to non-specific rare disease and special needs transition.
We have also included reference to such specificities in our discussion here:
‘The specifics of each rare disease or group of rare diseases contributes to unique transition issues. Furthermore, the rarity of experts in these rare pediatric and genetic dis-eases among adult health care professionals is a barrier to successful transition. Creative solutions are yet to be identified to overcome these barriers,’
Lines 69 to 74, Lines 272 to 275.
Comment 5:
Conclusion
The authors should keep their original idea and repeat it in the conclusion that any transition program should be implemented (or mandatory) in health systems.
Author response:
We have kept our original idea and repeated it in the conclusion that any transition program should be implemented in health systems.
Lines 281 to 282.
Round 2
Reviewer 1 Report
Thank you for incorporating the suggested corrections. The paper reads very well and now gives a clear perspecctive on the topic